# The Effects of Body Fat Reduction through the Metabolic Control of Steam-Processed Ginger Extract in High-Fat-Diet-Fed Mice

**DOI:** 10.3390/ijms25052982

**Published:** 2024-03-04

**Authors:** Yeong-Geun Lee, Sung Ryul Lee, Hyun Jin Baek, Jeong Eun Kwon, Nam-In Baek, Tong Ho Kang, Hyunggun Kim, Se Chan Kang

**Affiliations:** 1Department of Oriental Medicine and Biotechnology, Kyung Hee University, Yongin 17104, Republic of Korea; lyg629@nate.com (Y.-G.L.); guswls@nmr.kr (H.J.B.); jjung@nmr.kr (J.E.K.); nibaek@khu.ac.kr (N.-I.B.);; 2BioMedical Research Institute, Kyung Hee University, Yongin 17104, Republic of Korea; 3Department of Convergence Biomedical Science, Cardiovascular and Metabolic Disease Center, College of Medicine, Inje University, Busan 47392, Republic of Korea; lsr1113@inje.ac.kr; 4Department of Biomechatronic Engineering, Sungkyunkwan University, Suwon 16419, Republic of Korea

**Keywords:** anti-obesity, metabolic syndrome, steam process, *Zingiber officinale*

## Abstract

The prevalence of metabolic syndrome is increasing globally due to behavioral and environmental changes. There are many therapeutic agents available for the treatment of chronic metabolic diseases, such as obesity and diabetes, but the data on their efficacy and safety are lacking. Through a pilot study by our group, *Zingiber officinale* rhizomes used as a spice and functional food were selected as an anti-obesity candidate. In this study, steam-processed ginger extract (GGE) was used and we compared its efficacy at alleviating metabolic syndrome-related symptoms with that of conventional ginger extract (GE). Compared with GE, GGE (25–100 μg/mL) had an increased antioxidant capacity and α-glucosidase inhibitory activity in vitro. GGE was better at suppressing the differentiation of 3T3-L1 adipocytes and lipid accumulation in HepG2 cells and promoting glucose utilization in C2C12 cells than GE. In 16-week high-fat-diet (HFD)-fed mice, GGE (100 and 200 mg/kg) improved biochemical profiles, including lipid status and liver function, to a greater extent than GE (200 mg/kg). The supplementation of HFD-fed mice with GGE (200 mg/kg) resulted in the downregulation of SREBP-1c and FAS gene expression in the liver. Collectively, our results indicate that GGE is a promising therapeutic for the treatment of obesity and metabolic syndrome.

## 1. Introduction

Obesity contributes to metabolic syndrome [1], which is thought to be caused by an increase in adipose tissue mass following the proliferation of fat cells through adipogenesis [2] and which is strongly associated with the development of diabetes and fatty liver disease [3]. Non-alcoholic fatty liver disease (NAFLD) refers to the abnormal accumulation of fat in the liver and can progress to end-stage liver disease. Obesity, type 2 diabetes (T2DM), and NAFLD individually increase the risk of development of other diseases [3,4]. Drugs used to treat the lifestyle-related disorders of obesity, T2DM, and NAFLD have side effects and some individuals develop resistance to these drugs [1,3]. In addition to dietary and lifestyle modifications (i.e., restriction of calorie intake and increased physical activity) [1], functional foods can help prevent and treat obesity by improving lipid metabolism in the liver, controlling adipocyte differentiation, [5] and preserving normal skeletal muscle metabolism [6].

*Zingiber officinale* (commonly known as ginger) is a perennial herbaceous plant in the Zingiberaceae family. It is currently cultivated in India, China, Australia, Japan, and Korea. Underground stems or rhizomes of ginger are used for medicinal purposes. Ginger rhizomes are traditionally used as flavoring spices, seasonings, sliced snacks, beverages, and herbal ingredients for treating the common cold [7,8,9]. Among over 100 compounds reported in ginger, the major bioactive components in ginger rhizomes are essential volatile oils (e.g., terpenoids) and non-volatile pungent compounds (e.g., gingerols, shogaols, paradols, and zingerone) [10,11]. Like most vegetables, ginger contains a variety of vitamins and minerals. Ginger displays antioxidant, anti-inflammatory, anti-cancer, neuroprotective, anti-hyperglycemic, and anti-obesity effects [9,12,13]. In addition, ginger has been shown to be useful for the management of vomiting and nausea in pregnant women [14]. Ginger is “generally recognized as safe” (GRAS) by the US Food and Drug Administration. Although interactions between “natural” medicines like ginger and drugs are a major safety concern, few ginger–drug interactions have been reported [15,16].

Natural foods with bioactive components can help prevent lifestyle-related disorders [1,5]. To improve the usefulness of ginger as a functional food, several processing strategies, such as fermentation, steaming (or steam-drying), aging, and roasting, have been investigated [8]. Steaming is a key method used to process oriental herbal medicines to increase the content of bioactive compounds [8,17]. Recently, steaming procedures have been applied to ginger to enhance its functionality [18]. The health benefits have been found to vary depending on steaming conditions, such as the temperature and duration of steaming [8,18]. The dynamic conversion of ginger’s active components, especially 1-dehydro-6-gingerdione, following steaming and the differences in the ability of steamed ginger extract to suppress obesity and improve metabolic syndrome have not been thoroughly investigated [8,18,19,20,21]. Studies have reported that this compound has excellent antioxidant and anti-inflammatory effects [22,23]. In addition, our previous research confirmed that 1-dehydro-6-gingerdione and its crude extract, steam-processed ginger extract (GGE), have significant antidiabetic effects [24]. In this study, we would like to confirm the anti-obesity effect of GGE, which contains 1-dehydro-6-gingerdione as the primary component showing excellent effects on metabolic diseases. Hence, we investigated whether ginger extract prepared via steaming at a high temperature and high pressure (GGE) displayed a greater therapeutic potential for treating metabolic syndrome than ginger prepared using a conventional extraction method (GE). First, we determined the effects of GGE on antioxidant capacity and α-glucosidase inhibitory activity. Second, we examined the effects of GGE on adipocyte differentiation, glucose uptake, and lipid accumulation in an in vitro system. The excessive consumption of fat-rich diets is known to be a major risk factor for the development of metabolic syndromes, such as obesity and T2DM [4,22]. Finally, we investigated the effects of GGE on the alleviation of obesity and metabolic dysregulation in 16-week high-fat-diet (HFD)-fed mice. To perform this, body weight, organ weight, biochemical profiles, and the expression of several genes involved in metabolic regulation were assessed and compared among the treatment and control groups, and a histochemical analysis of the liver was performed.

## 2. Results

### 2.1. HPLC Quantitative Analyses of GE and GGE

To isolate the compound that contributed to the difference, GE was partitioned into *n*-hexane and H_2_O fractions. Then, repeated column chromatographies on the *n*-hexane fraction led to the isolation of 1-dehydro-6-gingerdione. Calibration curves for 1-dehydro-6-gingerdione were created using five concentrations (3.125 to 50 μg/mL). The quantifications of 1-dehydro-6-gingerdione in GE and GGE were performed from the areas of the peak recorded at 254 nm compared with the calibration curves created using standard solutions for the compound under analysis conditions. The regression curve and its correlation coefficient (r^2^) of 1-dehydro-6-gingerdione was evaluated (y = 4476.7x − 1445.9 (r^2^ = 1.000)). As a result, 1-dehydro-6-gingerdione was eluted at 46.53 min, and its amounts in GE and GGE were determined to be 0.19 ± 0.03 and 1.18 ± 0.15 mg/g, respectively (Figure 1).

### 2.2. Antioxidant Capacity of GGE

The steaming of ginger can affect its antioxidant capacity. We therefore determined the antioxidant capacities of GE and GGE using the ORAC assay (Table 1). GGE exhibited a higher antioxidant capacity than GE at the same concentration.

### 2.3. Inhibitory Effects of GGE on α-Glucosidase Activity

Next, we evaluated whether steaming improved the α-glucosidase inhibitory effect of ginger. As shown in Table 2, the α-glucosidase inhibitory effect of GGE is significantly higher than that of GE. The α-glucosidase inhibitory effects of equivalent doses of GE and GGE were less than 50%. The half-maximal inhibitory concentration of GGE was above 100 μg/mL and thus GGE is a mild α-glucosidase inhibitor. Based on the higher antioxidant capacity and α-glucosidase inhibitory activity of GGE compared to GE, steaming appears to augment the medicinal potential of ginger against hyperglycemia.

### 2.4. Inhibitory Effects of GE and GGE on Adipocyte Differentiation

The inhibition of adipogenesis can reduce lipid accumulation in the body, and therefore obesity. To evaluate the ability of GGE to inhibit adipocyte differentiation, 3T3-L1 cells were stimulated with insulin, dexamethasone, and isobutylmethyl xanthine (MDI) in the presence or absence of GE and GGE (25–100 μg/mL). As shown in Figure 2A, GE is not effective at inhibiting adipocyte differentiation. However, GGE at 50 and 100 μg/mL had a significant inhibitory effect on adipocyte differentiation (*p* < 0.01). These results suggest that GGE can suppress adipocyte differentiation in contrast to GE.

Therefore, we investigated the effect of GGE on transcript levels of C/EBPα, PPARγ, aP2, FAS, and CD36, which inhibited or promoted adipocyte differentiation in 3T3-L1 cells using qRT-PCR (Figure 2B). Differentiation stimulus resulted in a significant increase in levels of these five genes, which GE treatment was unable to suppress. In contrast, GGE treatment (100 μg/mL) significantly suppressed the mRNA levels of these five genes (*p* < 0.05). These results indicate that GGE treatment can suppress the transcription of adipogenesis-associated genes and adipocyte differentiation (Figure 2).

### 2.5. Inhibitory Effects of GGE on Lipid Accumulation in HepG2 Cells

We investigated the ability of GGE and GE to suppress lipid accumulation in HepG2 cells (Figure 3A). The exposure of HepG2 cells to 5% oleic acid resulted in the doubling of lipid accumulation (*p* < 0.05). GE at a given dose was not effective at suppressing lipid accumulation in HepG2 cells compared to the 5% oleic acid control. In contrast, treatment with GGE at 100 μg/mL resulted in a significant reduction in lipid accumulation (*p* < 0.01). Based on these results (Figure 3A), we examined the expressions of four genes related to intracellular fatty acid synthesis, namely LXR, SREBP, ACC, and FAS, using qRT-PCR (Figure 3B). In HepG2 cells, 5% oleic acid treatment increased the mRNA expression levels of SREBP, ACC, and FAS significantly (*p* < 0.05), but not that of LXR (Figure 3B). GE treatment at 100 μg/mL resulted in a remarkable decrease in the mRNA expression levels of SREBP and FAS. The oleic acid-induced increase in the mRNA expression levels of LXR, SREBP, ACC, and FAS genes was strongly suppressed by GGE in a dose-dependent manner. As shown in Figure 3B-(d), GGE at a concentration of 100 μg/mL significantly decreases the 5% oleic acid-induced increase in expression of the FAS gene (*p* < 0.05). This potent inhibitory effect of GGE on FAS gene expression can contribute to the suppression of lipid accumulation in HepG2 cells.

### 2.6. Stimulatory Effect of GGE on Glucose Uptake by C2C12 Cells

To evaluate the ability of GGE to improve the glucose uptake rate, C2C12 cells were fully differentiated, and then the uptake rates of 2-NDBG into cells were determined in the presence or absence of GE and GGE. Insulin treatment resulted in a significant increase in glucose uptake (Figure 4A). GE treatment at a given dose significantly increased glucose uptake in the absence of insulin (*p* < 0.01), but its potency was inversely related to the dose. GGE treatment increased the uptake of glucose significantly in a dose-dependent manner (*p* < 0.01). These results suggest that GGE stimulates glucose uptake more predictably than GE.

In skeletal muscles or adipocytes, the rate of glucose uptake is controlled by the actions of insulin receptor substrate 1 (IRS-1) and glucose transporter 4 (GLUT4). To identify genes possibly involved in the stimulation of glucose uptake (Figure 4B), we investigated the effect of GGE on the mRNA expression levels of GLUT4 and IRS genes. As shown in Figure 4B, insulin increases the mRNA expression levels of GLUT4 and IRS in C2C12 cells (*p* < 0.01). GGE treatment (100 μg/mL) results in a significant increase in the mRNA expression levels of GLUT4 (*p* < 0.05) and IRS (*p* < 0.01) compared with the untreated control. These results indicate that GGE can stimulate the uptake of glucose in part by the upregulation of GLUT4 and IRS genes.

### 2.7. Effects of GGE on Body Weight, Tissue Weight, Food Intake, and Water Intake

Considering GGE’s antioxidant capacity, inhibitory effect on α-glucosidase and adipocyte differentiation, and stimulation of glucose uptake, we further investigated the efficacy of GGE on alleviating obesity and NAFLD in an HFD-fed mouse model. GGE was dosed at 50, 100, and 200 mg/kg, whereas GE was dosed at 200 mg/kg to compare the effects of GGE and GE. During the 16-week experimental period, changes in the body weight, food consumption rate, and water consumption rate over time were evaluated and compared among groups. There was no significant difference in initial body weight among groups. However, after 16 weeks of experiments, the liver tissue weight of mice fed an HFD increased significantly compared to normal control mice (*p* < 0.05), but the kidney and spleen weights did not increase (*p* > 0.05). Liver weight was significantly decreased with GE (200 mg/kg) and GGE (50–200 mg/kg) treatments compared with the HFD-fed group (*p* < 0.01). A lower dose of GGE than GE had a similar suppressive effect on liver weight gain (Table 3). Neither GE nor GGE caused any significant changes in the weights of the kidney and spleen. Although GE and GGE were both effective at suppressing liver weight gain in the HFD condition, GGE had a more potent effect than GE.

The feeding efficiency ratio (FER) was calculated as a function of body weight change and dietary intake. As shown in Table 3, HFD-fed mice experienced significant increases in body weight gain, FER, and water consumption than the control group (*p* < 0.05). Among the HFD-fed groups, GGE treatment lowered the rate of food consumption and body weight gain as well as FER, but without statistical significance. The rate of water consumption was significantly decreased by GE (200 mg/kg) and GGE (100 and 200 mg/kg) treatments (Table 3).

### 2.8. Effects of GGE on HFD-Associated Changes in Biochemical Profiles

At the end of the 16-week experiment, biochemical profiles, including blood glucose, TG, cholesterol, and free fatty acid (NEFA) levels, were determined (Table 4). HFD-fed mice had significantly higher blood levels of total cholesterol (T-C), LDL-C, and NEFA (*p* < 0.05) than the control mice, but not glucose or TG levels (*p* > 0.05). GOT and GPT levels were significantly higher in HFD-fed mice than the control mice (each *p* < 0.05). Supplementation with GE (200 mg/kg) or GGE (50–100 mg/kg) resulted in a significant decrease in T-C (*p* < 0.05), LDL-C, and GPT levels compared with the HFD control group (*p* < 0.05). GGE had stronger suppressive effects than GE (Table 4).

### 2.9. Effects of GGE on Fat Accumulation and mRNA Expression Levels of Fat Accumulation-Related Genes in the Liver

An excessive intake of fat causes the accumulation of fat in the liver, leading to NAFLD. HFD-fed mice showed a significant increase in NEFA, T-C, and LDL-C levels in the blood (Table 4). We therefore performed oil-red O staining of cross-sections of liver tissue to investigate the effects of GGE on fat accumulation in the liver. As shown in Figure 5, HFD-fed mice showed a significant accumulation of fat in the liver compared with the normal-diet control mice (*p* < 0.05). GE supplementation at a dose of 200 mg/kg significantly reduced fat accumulation in the liver (*p* < 0.01). GGE supplementation also significantly suppressed fat accumulation in the liver in a dose-dependent manner (*p* < 0.01). The suppressive effect of GGE on fat accumulation in the liver is higher than that of GE because GGE at 50 mg/kg has a similar suppressive effect to that of GE at 200 mg/kg (Figure 5).

To further characterize the mechanism by which GE and GGE suppressed fat accumulation in the liver, the mRNA expression levels of SREBP-1c, PPAR-r, aP2, ACC, FAS, and CD36 were measured using qRT-PCR (Figure 6). The expression of these six genes in HFD-fed mice increased compared to the normal-diet control mice, but only the increase in transcript levels of SREBP-1c and aP2 was statistically significant (*p* < 0.05). GE supplementation (200 mg/kg) decreased the mRNA expression levels of these six genes, but without statistical significance. GGE supplementation decreased the mRNA levels of these six genes. In particular, GGE at 200 mg/kg caused a significant decrease in SREBP-1c, PPAR-γ, and FAS gene expression levels (Figure 6). These results suggest that GGE supplementation can suppress fat accumulation in the liver by downregulating the gene expression levels of SREBP-1c, PPAR-γ, and FAS. Fat accumulation in the liver can cause liver damage. As shown in Table 4, GOT and GPT levels are significantly increased along with excessive fat accumulation in the liver of HFD-fed mice (Figure 5). GGE supplementation had more potent suppressive effects on high-fat-associated liver damage and fat accumulation than GE.

### 2.10. Effects of GGE on the mRNA Expression Levels of Myostatin and MyoD

HFD-fed mice showed an increased mRNA expression of myostatin, whereas the expression of myoD was decreased but without statistical significance (Figure 7). GE supplementation (200 mg/kg) of HFD-fed mice did not correct the HFD-mediated changes in the expression of these two genes in muscle tissue. However, GGE supplementation reversed the HFD-mediated changes in the expression of these two genes. These effects of GGE were significant at a dose of 200 mg/kg.

## 3. Discussion

Ginger is available in various forms: fresh root ginger, bleached ginger, preserved ginger, dried ginger, and others [8]. Processing methods, including toasting, steaming, cooking, and fermentation, are used to increase the pharmacological benefits and reduce the toxicity or side effects of natural food products [8]. For example, red ginseng is prepared by steaming fresh ginseng and then drying it; the resultant product possesses greater pharmacological efficacy and is associated with better storage quality than fresh ginseng [25]. Likewise, in this study, it was confirmed that the content of 1-dehydro-6-gingerdione, a major component of this plant, increased dramatically in the steaming process of GGE. Oxidative stress interferes with the ingestion of glucose in the muscles and reduces insulin secretion by pancreatic *β*-cells, resulting in insulin resistance and ultimately leading to metabolic syndrome [26]. Similarly, hepatic lipid overload causes oxidative stress in the liver. Therefore, antioxidants play a significant role in preventing and/or managing metabolic diseases. GGE showed a greater antioxidant capacity than GE in this study, indicating that steaming did not result in the loss of the biological activities of the ginger extract (Table 1). The GGE developed through this study confirmed that the concentration of 1-dehydro-6-gingerdione significantly increased compared to existing GE (Figure 1). The cause of the increasing antioxidant capacity was revealed as the “Keto-Enol Form” of a major compound in the GGE, 1-dehydro-6-gingerdione, which is produced during the steaming process [27,28]. Furthermore, 1-dehydro-6-gingerdione, identified as a major component in this study, is present in very low amounts in conventional ginger, which has limited its research to date. Previous studies on the anti-obesity inhibitory activity of ginger mainly focused on the major compound 6-gingerol [29,30,31,32,33,34]. Additionally, active components, such as 6-shogaol [35], gingerenone A [36], gingerenone [37], and galanolactone [38], have been found in the relevant literature. On the other hand, 1-dehydro-6-gingerdione has demonstrated an excellent anti-inflammatory activity, particularly surpassing the effectiveness of primary ginger compounds (6-shogaol, 6-dehydroshogaol, and hexahydrocurcumin) in terms of anti-inflammatory efficacy [22,23,39]. α-glucosidase is a key enzyme in carbohydrate metabolism. α-glucosidase inhibitors, such as acarbose, can suppress postprandial hyperglycemia. α-glucosidase inhibitors function as antidiabetic and anti-obesity agents in addition to reducing postprandial blood glucose levels [40]. GGE displayed α-glucosidase inhibitory potential and its activity was superior to that of GE (Table 2). Our research team recently reported the suppression of diabetes activity using GGE and, for the first time, demonstrated its anti-obesity suppression activity [25].

In healthy individuals, the skeletal muscle is the major site for postprandial glucose uptake, and an impairment of this process contributes to the pathogenesis of type 2 diabetes mellitus (T2DM). When insulin resistance occurs, insulin cannot act on skeletal muscle cells and glucose in the blood is not taken up by the muscle cells. Increasing the glucose uptake by skeletal muscle cells can help alleviate insulin resistance [6]. Thus, measuring glucose uptake by cultured myotubes is a reliable method to assess whether an intervention can improve insulin responsiveness or correct insulin resistance [41].

GGE treatment increases the mRNA expression levels of IRS-1 and GLUT4 in C2C12 cells to a greater extent than GE and increases 2-NDBG uptake more than GE (Figure 4). Insulin resistance due to obesity or diabetes causes muscle atrophy, which also affects circulating levels of glucose. Moreover, diabetic and obese conditions alter the structural, metabolic, and functional characteristics of skeletal muscle fibers, leading to the loss of muscles [42]. Especially, myostatin acts as a negative regulator of skeletal muscle mass and frequently increases in obesity. In contrast, MyoD plays a critical role in myogenesis. HFD feeding decreased the mRNA expression levels of MyoD and increased levels of myostatin. These changes indicate impaired muscle regeneration and, thus, a loss of muscle mass. Our results indicate that GGE supplementation is better than GE supplementation at protecting the body against the loss of muscle mass. Previous in vivo studies reported the effectiveness of ginger at doses of 200–500 mg/kg of body weight [43,44,45,46,47]. However, the steamed ginger extract developed in our study, abbreviated as GGE, demonstrated superior efficacy compared to the previous ginger extract (GE) at concentrations in the range of 100–200 mg/kg of body weight.

While HFD-induced obese mice showed increased body weight, food consumption, and water consumption results than the control mice, GGE supplementation decreased body weight gain and water consumption rate with a higher potency than GE. In addition, the accumulation of free fatty acids and TG in the liver is a key feature of NAFLD and is often observed in insulin resistance due to obesity. The inhibition of lipid accumulation in the liver can help treat non-alcoholic fatty liver. In general, when abnormal lipid metabolism occurs in the liver due to the ingestion of fat, the weight of the liver increases with lipid deposition, and lipid and cholesterol levels in the liver increase, possibly resulting in the development of NAFLD in the context of obesity [48]. Adipogenesis (adipocyte differentiation) is controlled by a transcriptional network coordinated by numerous transcription factors, including C/EBPα and PPARγ, which inhibit or promote adipocyte differentiation [2]. Mechanistically, an increase in body weight is associated with the differentiation of pre-adipocyte cells, and adipocyte differentiation is closely related to insulin resistance [2,4]. Therefore, the inhibition of excessive adipocyte differentiation is important to prevent an increase in body weight. GGE had a greater inhibitory effect on adipocyte differentiation than GE (Figure 2A). In addition to suppressing lipid accumulation in HepG2 cells and the livers of HFD-fed mice, GGE downregulated the expression of genes involved in fatty acid synthesis (e.g., SREBP, PPAR-γ, and FAS) in the liver to a greater extent than GE. HFD feeding also resulted in aberrant levels of total cholesterol, LDL-C, GOT, GPT, and NEFA, but GGE supplementation reduced these aberrant changes with a higher potency than GE. In a clinical context, interventional approaches to prevent obesity and/or metabolic syndrome can take an extended period of time and therefore cause gastrointestinal problems when used as pills. A recent study reported that steamed ginger extract exhibited antiulcer activity against EtOH/HCl mixture-induced gastric damage in rats [21]. This suggests that the long-term use of ginger extract in the form of a pill to treat obesity and/or metabolic syndrome is not associated with a high risk of gastrointestinal complications.

Our study had several limitations. Firstly, only male mice were included in the experiment to exclude the complex effects of hormones on lipid metabolism. Future studies should include female groups to confirm our findings. Secondly, we prepared the GGE by steaming fresh ginger and found that the efficacy of the GGE was superior to that of GE based on a treatment dose. However, it is unclear whether the steaming process used in this study is the best processing option and various steaming conditions with variations in temperature, pressure, and duration should be evaluated to assess the best steaming process for ginger. Finally, the main active compounds that differ between GE and GGE should be identified to improve our understanding of the underlying anti-obesity mechanisms of GGE.

## 4. Materials and Methods

### 4.1. Chemicals and Reagents

Dulbecco’s modified Eagle’s medium (DMEM) was purchased from Gibco (Grand Island, NY, USA). Bovine calf serum (BCS), fetal bovine serum (FBS), penicillin (100 units/mL)/streptomycin (100 μg/mL), and Trizol were purchased from Invitrogen (Carlsbad, CA, USA). Acarbose, dexamethasone, fluorescein, insulin, oleic acid, Trolox, 3-isobutyl-1-methylxanthine, *α*-glucosidase, and *β*-phycoerythrin were obtained from Sigma Aldrich (St. Louis, MO, USA). AdipoRed^TM^ assay reagent was purchased from Lonza (Basel, Switzerland). 2-[*N*-(7-nitrobenz-2-oxa-1, 3-diazol-4-yl)amino]-2-deoxy-D-glucose (2-NBDG) was obtained from Thermo Fisher Scientific (Carlsbad, CA, USA). Chemicals, unless otherwise stated, were purchased from Sigma Aldrich.

### 4.2. Preparation of GE and GGE

The ginger extract (GE) and golden ginger extract (GGE) we used were the same as those stated in our previously published papers [24]. In brief, *Z. officinale* Roscoe was purchased from a local market (Wanju-gun, Republic of Korea) in April 2019 and identified by Professor Se Chan Kang (Kyung Hee University, Yongin, Republic of Korea). A voucher specimen (No. BMRI2019-1) was deposited in the Laboratory of Natural Medicine Resources at the BioMedical Research Institute, Kyung Hee University. Ginger was washed three times with distilled water to remove sand and dust. Golden ginger (GG) was prepared by steaming using the following conditions: 2.0~2.5 kgf/m^2^, 97 °C, 2 h. After drying for 30 h at 50 °C in a dry oven, the steamed ginger was recovered for further extraction. GGE was obtained by extracting the GG with a 15-fold volume of 70% ethanol (*v*/*v*) for 15 h at 85 °C. Conventional ginger extract (GE) was prepared by extracting washed fresh ginger a 15-fold volume of 70% ethanol (*v*/*v*) for 15 h at 85 °C. GGE and GE were filtered and concentrated, and the extracts were then spray-dried using a rotary vacuum evaporator (EYELA, Tokyo Rikakikai Co., Tokyo, Japan) to obtain powder and stored at −20 °C until use. Both the GGE and GE were dissolved in distilled water and 0.85% saline for in vitro and animal studies, respectively.

### 4.3. HPLC Quantitative Analyses of GE and GGE

To validate and determine the differences in the chemical components between GGE and GE, a high-performance liquid chromatography (HPLC) analysis was conducted. Briefly, powdered GE and GGE samples were dissolved in 80% methanol to obtain 10,000 ppm solutions. Both solutions were passed through a 0.22 μm membrane filter (Woongki Science Co., Ltd., Seoul, Republic of Korea); then, 10 μL of the filtrate was injected into a Waters 600S HPLC system (Waters, Milford, MA, USA) equipped with a Waters 2487 UV detector (280 nm). A 250 × 4.6 mm Shimpack Gist column with a particle size of 3 μm was used (Shimadzu Co., Kyoto, Japan). The mobile phase (0.1% formic acid in H_2_O, solvent A; acetonitrile, solvent B) was eluted with the following elution gradient of B: 30% (0.01 min) → 30% (5 min) → 55% (10 min) → 55% (13 min) → 80% (25 min) → 80% (30 min) → 100% (60 min). Triplicate samples were quantified using instrument-embedded software.

To isolate the compound that contributed to the difference in the HPLC chromatogram, column chromatography (c.c.) was performed. The dried rhizomes of *Z. officinale* (20.0 kg) were extracted with 70% aqueous EtOH (90 L × 4) at room temp. for 24 h. After filtration and concentration, the obtained concentrated EtOH extract (ZOE, 1.6 kg) was poured into water (4.0 L) and successively partitioned with *n*-hexane (4.0 L × 3). Each layer was concentrated under reduced-pressure conditions to obtain the *n*-hexane (ZOH, 539 g), water (ZOW, 1061 g), and residue (ZOHR, 326 g). Fraction ZOHR (326 g) was subjected to SiO_2_ column chromatography (c.c., Φ 7.0 × 16.0 cm) and eluted with *n*-hexane-EtOAc (4:1 → 1:1, 500 mL of each) to CHCl_3_-MeOH-H_2_O (30:3:1, 500 mL of each), with monitoring by TLC, yielding 14 fractions (ZOHR-1 to ZOHR-14). ZOHR-2 (46.3 g, Ve/Vt 0.125–0.250) was subjected to ODS c.c. (Φ 13 × 6 cm, acetone-water = 1:1, 8 L) to yield 11 fractions (ZOHR-2-1 to ZOHR-2-11). ZOHR-2-6 (4.7 g, Ve/Vt 0.430–0.470) was subjected to SiO_2_ c.c. (Φ 2 × 15 cm, CHCl_3_-EtOAc = 50:1 → 30:1 → 10:1, 2.7 L of each) to yield 16 fractions (ZOHR-2-6-1 to ZOHR-2-6-16) along with purified compound 1 (ZOHR-2-6-9, 100.8 mg, Ve/Vt 0.517–0.535, TLC [SiO_2_] Rf 0.58, *n*-hexane-EtOAc = 3:1, TLC [ODS] Rf 0.51, acetone-MeOH-water = 4:1:1).

1-dehydro-6-gingerdione: yellow oil (CHCl_3_); C_17_H_22_O_4_; EI-MS *m*/*z* 290 [M]; ^1^H-NMR (600 MHz, CDCl_3_, *δ*_H_) 0.90 (3H, t, *J* = 6.8 Hz), 1.33 (4H, m), 1.61 (2H, m), 2.39 (2H, t, *J* = 7.8 Hz), 3.89 (3H, s), 5.63 (1H, s), 6.35 (1H, d, *J* = 15.6 Hz), 6.91 (1H, d, *J* = 8.4 Hz), 7.02 (1H, d, *J* = 1.8 Hz), 7.08 (1H, dd, *J* = 8.4, 1.8 Hz), 7.52 (1H, d, *J* = 15.6 Hz); ^13^C-NMR (150 MHz, CDCl_3_, *δ*_C_) 200.19, 178.03, 147.64, 146.76, 139.81, 127.70, 122.59, 120.54, 114.79, 109.44, 100.12, 55.91, 40.08, 31.45, 25.30, 22.43, 13.91.

### 4.4. Cell Lines

3T3-L1 mouse fibroblast cells, C2C12 mouse myoblast cells, and HepG2 human hepatocyte cells were obtained from the American Type Culture Collection (ATCC, Manassas, VA, USA). 3T3-L1 cells were grown in DMEM supplemented with 10% BCS and 1% penicillin/streptomycin. C2C12 cells and HepG2 cells were grown in DMEM supplemented with 10% FBS and 1% penicillin/streptomycin. Cells were maintained in a humidified incubator at 37 °C and 5% CO_2_.

### 4.5. ORAC Assay

To determine the antioxidant capacity of GGE, the oxygen radical absorbance capacity (ORAC) assay, which measures radical scavenging activity against peroxyl radicals induced by 2,2′-azobis dihydrochloride (AAPH), was performed. The loss of the fluorescence of fluorescein (FL) indicated the extent of the reaction with peroxyl radicals. Briefly, 2 µL of the sample or Trolox was incubated with 0.2 mM of *β*-phycoerythrin and 200 mM of AAPH in a total volume of 200 µL. The decrease in fluorescence was determined at 2-minute intervals for 60 min at 37 °C. All ORAC analyses were performed on a Synergy HT plate reader (BioTek Instruments Inc., Winooski, VT, USA) at 37 °C with an excitation wavelength of 535 nm and an emission wavelength of 590 nm. The antioxidant effects of GE and GGE were measured by comparison with a known antioxidant, Trolox, which is a water-soluble analog of vitamin E. The area under the fluorescence decay curve for FL (AUC) was calculated as follows: AUC (area under the curve) = 1 + f1/f0 + f2/f0 + f3/f0 + … + f19/f0 + f20/f0. The ORAC value was calculated as follows: [(AUC_sample_ − AUC_blank_)/(AUC_Trolox_ − AUC_blank_)] × (molarity of Trolox/molarity of sample). Final ORAC values were expressed as mean ± SEM.

### 4.6. α-Glucosidase Inhibition Assay

The α-glucosidase inhibition assay was performed with an ELISA reader (TECAN, Basel, Switzerland) at a wavelength of 405 nm according to a previous study [44]. Acarbose was used as a specific inhibitor of α-glucosidase. Briefly, 10 μL of GE and GGE at given doses was added to a reaction mixture containing 79 μL of sodium phosphate buffer (0.1 M, pH 6.8), 10 μL *p*-nitrophenyl glucopyranoside (50 mM), and 1 μL *α*-glucosidase (0.5 U/mL) at 37 °C for 30 min. After adding 200 μL of stop buffer (2 M NaOH), the optical density was measured using an ELISA plate reader at a wavelength of 405 nm. Inhibition was calculated as follows:α-glucosidase inhibition (%) = (Control_Absorbance_ − Sample_Absorbance_)/Control_Absorbance_ × 100

### 4.7. Adipocyte Differentiation Assay

As described previously [49], 3T3-L1 preadipocytes were seeded onto 96-well plates (1 × 10^4^ cells/well). To induce differentiation, 2-day post-confluent 3T3-L1 cells (designated day 0) were fed DMEM containing 10% FBS, 1 μM dexamethasone, 0.5 mM 3-isobutyl-1-methylxanthine, and insulin for 2 days. The cells were then cultured further in DMEM supplemented with 10% FBS every other day in the presence of GE and GGE (25, 50, and 100 μg/mL). On day 7, AdipoRed staining was performed to assess the differentiation into adipocytes and to determine the rate of adipocyte differentiation. Briefly, the cells were washed twice with phosphate-buffered saline (PBS, pH 7.2), fixed in 4% formaldehyde (St. Louis, MO, USA) at room temperature for 4 h, and stained with AdipoRed^TM^ assay reagent for 10 min. The plates were placed in a fluorometer (TECAN, Switzerland), and fluorescence was measured using excitation and emission wavelengths of 485 nm and 535 nm, respectively.

### 4.8. Oleic Acid-Induced Lipid Accumulation Assay

As previously described [50], HepG2 cells were seeded on 96-well plates (5 × 10^3^ cells/well). To induce lipid accumulation, HepG2 cells were cultured in DMEM containing 20% oleic acid for 3 days, after which the medium was replaced with DMEM containing 20% oleic acid every day. During the induction of lipid accumulation in the HepG2 cells, the cells were treated with GE and GGE at concentrations of 25, 50, and 100 μg/mL in the culture medium. At the end of the experiment, AdipoRed staining was performed to assess lipid accumulation by measuring fluorescence absorbance. Briefly, the cells were washed twice with PBS (pH 7.2), fixed in 4% formaldehyde at room temperature for 4 h, and stained with AdipoRed^TM^ assay reagent for 10 min. Plates were placed in a fluorometer, and the fluorescence was measured at an excitation wavelength of 485 nm and emission wavelength of 535 nm.

### 4.9. 2-NBDG Uptake Assay

Glucose uptake activity was analyzed by measuring the uptake of 2-NBDG, a fluorescent D-glucose indicator, as described previously [50]. Briefly, C2C12 cells were seeded in 96-well plates (1 × 10^3^ cells/well) and cultured to 70% confluence in DMEM. Then, the media was switched to myoblast differentiation media containing DMEM supplemented with 2% horse serum, which was replaced every 2 days. Fully differentiated cells were incubated in serum-free medium for 24 h and then treated with 2-NBDG (50 nM) in the presence or absence of GE or GGE for 24 h. Insulin (100 nM) was used as a positive control for the glucose uptake assay. After washing the cells three times with ice-cold PBS to stop the reaction, the intracellular uptake of 2-NBDG was measured using a fluorometer at excitation and emission wavelengths of 485 and 535 nm, respectively.

### 4.10. Quantitative RT-PCR Analysis

Total RNA from each sample was extracted using a TRIzol reagent (Invitrogen, USA) according to the manufacturer’s protocol. The quantity and purity of the RNA were calculated by measuring absorbance values at 260 nm and 280 nm using a NanoDrop instrument (Thermo Scientific, Waltham, MA, USA). Reverse transcription was performed with 0.5 μg of total RNA to generate double-stranded complementary DNA using a PrimeScript^TM^ II 1st Strand cDNA Synthesis Kit (Takara, Tokyo, Japan), and quantitative real-time PCR (qRT-PCR) was performed using an MX3005P thermocycler (Stratagene, San Diego, CA, USA). Primer sequences used for qRT-PCR are shown in Table 5. The final OCR reaction volume was 25 μL comprising 2 μL of cDNA template, 12.5 μL of Master Mix, 1 μL of each primer (10 μM stock solution), 8.5 μL of sterile distilled water, and SYBR Premix Ex Taq II (Takara, Japan). The thermal cycling profile consisted of a pre-incubation step at 95 °C for 10 min followed by 40 cycles of 95 °C for 15 s and 60 °C for 60 s. Relative mRNA levels were determined using the comparative cycle threshold method. Results were normalized by the mRNA level of *β*-actin and expressed as a ratio relative to the untreated control.

### 4.11. Animals and Study Design

All animals received humane care. The experimental animal facility and study protocols (KHUASP(GC)-17-029) were approved by the Animal Care and Use Committee. All experimental procedures were undertaken in compliance with the Guide for the Care and Use of Laboratory Animals (National Institutes of Health, USA) and the National Animal Welfare Law of the Republic of Korea. C57BL/6 male mice (4 weeks old; 13–18 g body weight) were obtained from Central Lab Animal Inc. (Seoul, Republic of Korea). The mice were maintained in a controlled environment of 22 ± 1 °C and humidity of 50 ± 10% with a 12 h light/dark cycle and provided with tap water every day. After a one-week acclimation period, the mice were housed separately in cages and familiarized with the testing procedures.

The normal-control diet group was fed a chow diet (2018S Teklad Global 18% Protein Rodent diet; Envigo, Madison, WI, USA) for 16 weeks. We used an atherogenic diet (ATH diet, D12336; Research Diets, Inc., New Brunswick, NJ, USA) as the high-fat diet (HFD). The GE dose was set at 200 mg/kg of body weight. Three different doses of GGE—50, 100, and 200 mg GG/kg of body weight—were used. The mice were fed a HFD for 4 weeks to induce obesity and then randomly divided into five groups (n = 6 each group): HFD control, GE group, and three GGE groups (three different GG doses). An appropriate dosing volume of saline or extract was determined after daily weighing. Normal saline, GE, or GG were delivered daily by oral gavage. The intragastric delivery of saline or extract was carefully performed by a well-trained researcher to minimize animal stress. During the 16-week experiment, weight gain as well as food and water intakes were calculated by collecting and weighing uneaten food and water twice per week.

### 4.12. Biochemical Analysis and Determination of Tissue Weight

At the end of the 16-week experiment, all animals were fasted for 12 h, and blood was collected through the abdominal vena cava under anesthesia with diethyl ether. The blood was allowed to clot for 30 min. Serum was then separated by centrifugation at 3500× *g* for 10 min at room temperature. Serum chemistry analyses of total cholesterol (T-Chol), low-density lipoprotein cholesterol (LDL-C), high-density lipoprotein cholesterol (HDL-C), triglyceride (TG), glucose, aspartate aminotransferase (GOT), and alanine aminotransferase (GPT) levels were performed using an AU480 Chemistry Analyzer with reagents supplied by the manufacturer (Beckman Coulter, Inc., Brea, CA, USA). After the mice were sacrificed, the liver, kidney and spleen were extracted, cleaned with a normal saline solution, blotted dry with filter paper, and then weighed.

### 4.13. Histological Analysis of the Liver

Liver tissues were fixed with 10% neutral formalin, and histology sections of a 4 μm thickness were prepared. Slides were then stained with 0.5% Oil-red O for 10 min. Sections were visualized under a BX51 microscope (OLYMPUS, Tokyo, Japan), and digital images were captured and analyzed using Image J software, version 1.53 (National Institute of Health, Rockville, MD, USA).

### 4.14. Statistical Analysis

Data are expressed as mean ± SEM (standard error of the mean; in vitro experiments) or mean ± SD (standard deviation; in vivo experiments). Significance was determined by one-way analysis of variance (ANOVA) followed by a modified *t*-test with Bonferroni corrections for comparisons between individual groups using SPSS 12 (SPSS Inc., Chicago, IL, USA). *p*-values less than 0.05 were considered statistically significant.

## 5. Conclusions

The content of 1-dehydro-6-gingerdione of GGE, which was prepared by steaming ginger at a high pressure, increased dramatically in the steaming process. GGE showed a greater antioxidant capacity, which plays a key role in preventing and/or managing metabolic diseases, than GE in this study, indicating that steaming did not result in the loss of the biological activities of the ginger extracts. We demonstrated that GGE was more effective than GE at alleviating obesity and metabolic symptoms caused by a 16-week high-fat diet. In an in vitro system, GGE displayed greater antioxidant capacity and α-glucosidase inhibitory activity than GE. GGE was also superior at suppressing the adipocyte differentiation of 3T3-L1 cells, promoting glucose utilization by C2C12 cells, and suppressing lipid accumulation in HepG2 cells than GE. These effects of GGE were associated with changes in the transcript levels of associated genes. In HFD-fed mice, GGE treatment improved biochemical profiles, including lipid status and liver function, with higher potency than GE. In addition, supplementation with GGE counteracted the impairment of skeletal muscles caused by a high-fat diet. In conclusion, GGE has great potential as a promising therapeutic for treating obesity and metabolic syndrome.

## Figures and Tables

**Figure 1 ijms-25-02982-f001:**
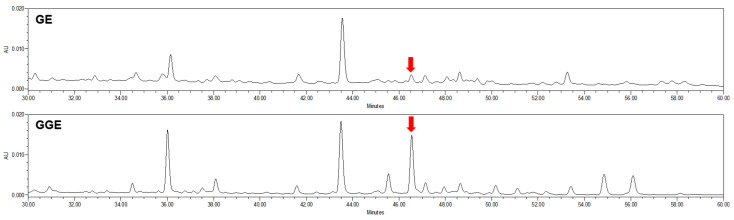
Representative HPLC chromatogram of ginger extract (GE) and steam-processed ginger extract (GGE). Red arrow, 1-dehydro-6-gingerdione.

**Figure 2 ijms-25-02982-f002:**
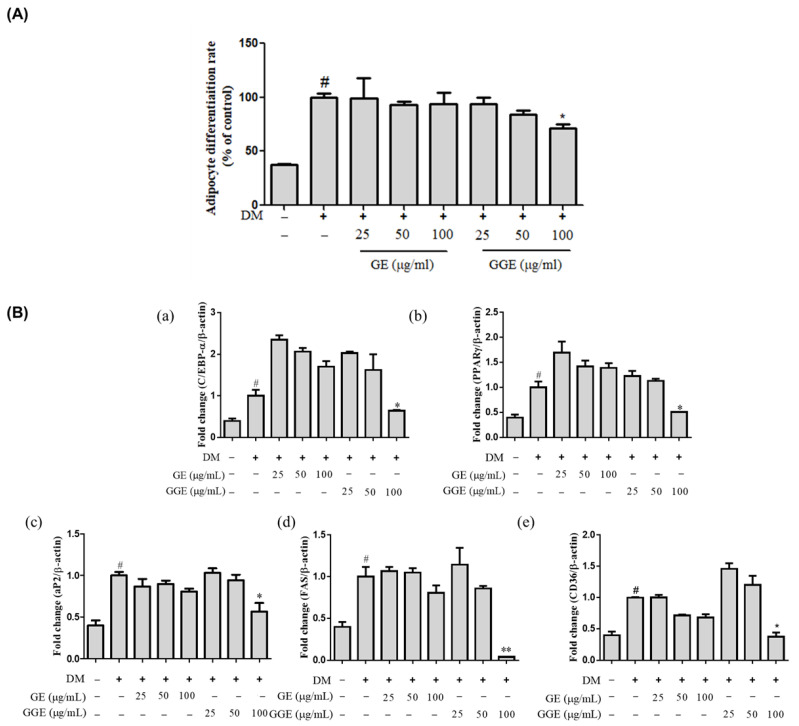
Inhibitory effects of GE and GGE on adipocyte differentiation. (**A**) Adipocyte differentiation inhibition rates (%) of GE and GGE in 3T3-L1 cells. (**B**) Inhibitory effects of GE and GGE on the mRNA expression levels of adipogenesis-associated genes. (**a**) C/EBP-α, (**b**) PPARγ, (**c**) aP2, (**d**) FAS, and (**e**) CD36. Data are expressed as a percentage of differentiation control and are mean ± SD (n = 4). ^#^
*p* < 0.05 vs. untreated control. * *p* < 0.05, ** *p* < 0.01 vs. differentiation control. aP2, adipocyte protein 2; C/EBP-α, CCAAT/enhancer binding protein; CD36, cluster of differentiation 36; DM, adipocyte differentiation medium; FAS, fatty acid synthase; GE, EtOH extract of fresh ginger; GGE, EtOH extract of golden ginger, which is steam-processed ginger; PPARγ, peroxisome proliferator activated receptor γ.

**Figure 3 ijms-25-02982-f003:**
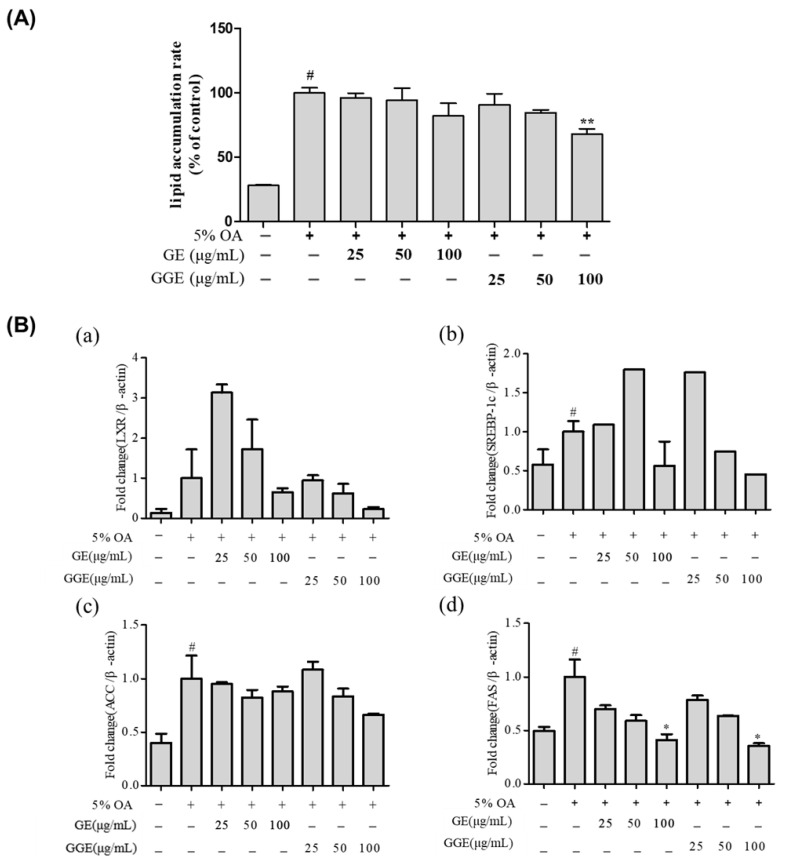
Inhibitory effects of GE and GGE on lipid accumulation in HepG2 cells. (**A**) Inhibitory effects of GE and GGE on intracellular lipid accumulation. (**B**) Inhibitory effects of GE and GGE on mRNA expression levels of lipogenesis-associated genes. (**a**) LXR, (**b**) SREBP-1c, (**c**) ACC, and (**d**) FAS. Data are expressed as a percentage of differentiation control and are the mean ± SD (n = 4). ^#^ *p* < 0.05 vs. untreated control, * *p* < 0.05 , ** *p* < 0.01 vs. 5% oleic acid-treated control; ACC, acetyl CoA carboxylase; FAS, fatty acid synthase; GE, EtOH extract of fresh ginger; GGE, EtOH extract of golden ginger, which is steam-processed ginger; LXR, liver X receptor; SREBP-1c, sterol regulatory element-binding protein-1c.

**Figure 4 ijms-25-02982-f004:**
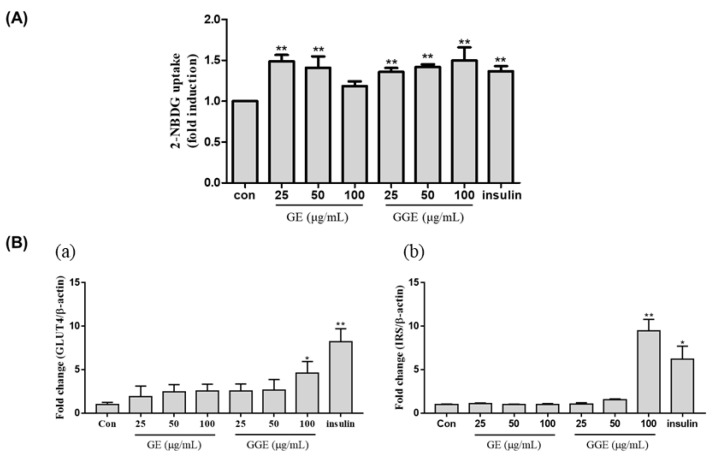
Stimulatory effects of GE and GGE on glucose uptake by C2C12 cells. (**A**) Stimulatory effects of GE and GGE on glucose uptake. (**B**) Stimulatory effects of GE and GGE on mRNA expression levels of GLUT4 and IRS genes. (**a**) GLUT4 and (**b**) IRS. Data are mean ± SD (n = 4). * and ** indicate *p* < 0.05 and *p* < 0.01 vs. untreated control, respectively. 2-NBDG, 2-[N-(nitrobenz-2-oxa-1,3-diazol-4-yl) amino]-2-deoxy-D-glucose; GLUT4, glucose transporter type-4; GE, EtOH extract of fresh ginger; GGE, EtOH extract of golden ginger, which is steam-processed ginger; IRS, insulin receptor substrate.

**Figure 5 ijms-25-02982-f005:**
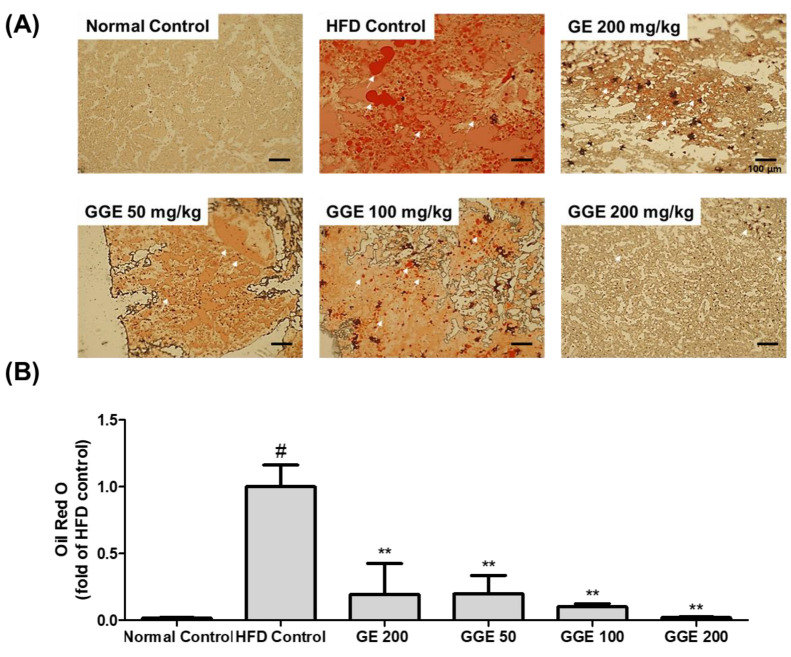
Inhibitory effects of GE and GGE on fat accumulation in the liver. (**A**) A representative image. (**B**) Quantification of lipids in the liver. Data are expressed as a percentage of HFD and are the mean ± SEM (each group, n = 6). ^#^ *p* < 0.05 vs. normal chow diet; ** *p* < 0.01 vs. HFD control; HFD, high-fat diet, GE, EtOH extract of fresh ginger; GGE, EtOH extract of golden ginger, which is steam-processed ginger.

**Figure 6 ijms-25-02982-f006:**
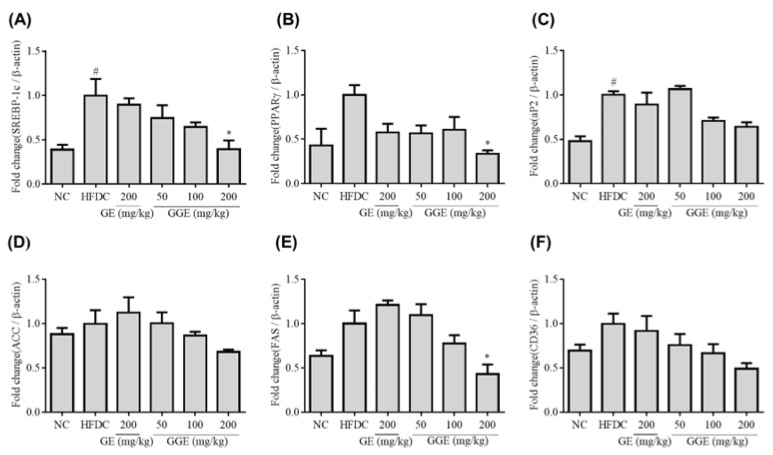
Inhibitory effects of GE and GGE on mRNA expression levels of lipid accumulation-associated genes in the liver. At the end of animal study, the mRNA expression levels of SREBP-1c (**A**), PPAR γ (**B**), aP2 (**C**), ACC (**D**), FAS (**E**), and CD36 (**F**) genes in the liver were determined by qRT-PCR. Results were normalized by the mRNA expression level of β-actin and expressed as a relative ratio to HFD control. Data are mean ± SD (each group, n = 6). * and ^#^ indicate *p* < 0.05 vs. NC group and HFD control group, respectively. ACC, acetyl CoA carboxylase; FAS, fatty acid synthase; HFDC, high-fat-diet control; GE, EtOH extract of fresh ginger; GGE, EtOH extract of golden ginger, which is steam-processed ginger; LXR, liver X receptor; NC; normal chow diet; SREBP-1c, sterol regulatory element-binding protein-1c.

**Figure 7 ijms-25-02982-f007:**
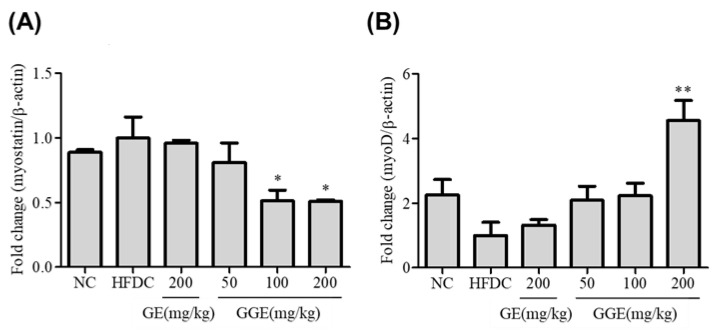
Effects of GE and GGE on mRNA expression levels of myostatin and myoD gene. At the end of animal study, the mRNA expression levels of myostatin (**A**) and myoD (**B**) gene in the muscle were determined by qRT-PCR. Results were normalized by the mRNA expression level of β-actin and expressed as a relative ratio to the HFD control. Data are mean ± SD (each group, n = 6). * and ** indicate *p* < 0.05 and *p* < 0.01 vs. HFD control, respectively. HFD, high-fat diet, GE, EtOH extract of fresh ginger; GGE, EtOH extract of golden ginger, which is steam-processed ginger.

**Table 1 ijms-25-02982-t001:** Antioxidant capacities of GE and GGE.

Dose (μg/mL)	Trolox Equivalent Antioxidant Capacity	
	GE	GGE
100	0.68 ± 0.006	0.80 ± 0.048
50	0.69 ± 0.066	0.78 ± 0.006
25	0.67 ± 0.057	0.76 ± 0.0017

The value of Trolox was set at 1.0 and the results are expressed as an index of Trolox. Data are mean ± SD (n = 4). GE, EtOH extract of fresh ginger; GGE, EtOH extract of golden ginger, which is steam-processed ginger.

**Table 2 ijms-25-02982-t002:** Inhibitory effects of GE and GGE on α-glucosidase activity (%).

Dose (μg/mL)	α-Glucosidase Inhibition Rate (%)		
	GE	GGE	Acarbose
100	11.92 ± 3.271 *	31.84 ± 4.403 **	53.33 ± 1.523 **
50	3.04 ± 5.444	31.51 ± 1.839 **^,##^	44.66 ± 0.239 **
25	0 ± 10.912	26.08 ± 4.557 **^,##^	35.36 ± 15.952 **
0	0 ± 0.58		

Acarbose was used as a positive control. Data are mean ± SD (n = 4) and are expressed as a percentage of control (0 μg/mL). * *p* < 0.05, ** *p* < 0.001 vs. 0 μg/mL. ^##^
*p* < 0.001 compared with the same concentration of GE. GE, EtOH extract of fresh ginger; GGE, EtOH extract of golden ginger, which is steam-processed ginger.

**Table 3 ijms-25-02982-t003:** Changes in body weight gain, tissue weight, food consumption, food efficiency ratio, and water consumption.

Group	Body Weight Gain (g/Day)	Relative Tissue Weight			Food Consumption (g/Day)	FER ^†^	Water Consumption (mL/Week)
		Liver	Kidney	Spleen			
C57BL/6 mice (each group, n = 6)							
Normal control	6.7 ± 1.72	3.7 ± 0.1	1.3 ± 0.1	0.3 ± 0.01	140.7 ± 11.46	4.7 ± 1.22	76.3 ± 4.73
HFD control	20.0 ± 1.39 ^#^	5.3 ± 0.2 ^#^	0.8 ± 0.03 ^#^	0.2 ± 0.01	130.1 ± 7.84	15.4 ± 1.07 ^#^	124.6 ± 25.48 ^#^
GE (mg/kg body weight/day)							
200	18.3 ± 0.96	2.7 ± 0.2 **	0.6 ± 0.04	0.2 ± 0.02	105.4 ± 9.59	17.3 ± 0.91	80.0 ± 2.88 *
GGE (mg/kg body weight/day)							
50	14.4 ± 1.61	2.8 ± 0.2 **	0.8 ± 0.03	0.2 ± 0.01	104.2 ± 10.10	13.8 ± 1.54	77.5 ± 4.28
100	14.2 ± 1.08	2.7 ± 0.2 **	0.7 ± 0.02	0.2 ± 0.01	116.1 ± 9.55	12.2 ± 0.93	84.2 ± 4.78 *
200	14.3 ± 1.87	3.2 ± 0.5 **	0.9 ± 0.02	0.2 ± 0.03	109.8 ± 9.27	13.0 ± 1.70	80.0 ± 4.00 *

^†^ FER (food efficiency ratio) = weight gain/food consumption × 100. Data are mean ± SD (each group, n = 6). ^#^ *p* < 0.05 vs. normal control, * *p* < 0.05, ** *p* < 0.01 vs. HFD control. HFD, high-fat diet; GE, EtOH extract of fresh ginger; GGE, EtOH extract of golden ginger, which is steam-processed ginger.

**Table 4 ijms-25-02982-t004:** Biochemical profiles of the HFD-induced obesity mice.

Group	Glucose (mg/dL)	Triglycerides (mg/dL)	Total Cholesterol (mg/dL)	HDL-C (mg/dL)	LDL-C (mg/dL)	GOP (U/L)	GPT (U/L)	NEFA (mEq/L)
C57 mice (n = 6 each)								
Normal control	43.0 ± 11.27	32.7 ± 8.33	59.3 ± 6.03	46.0 ± 1.73	8.0 ± 1.00	141.0 ± 18.73	71.0 ± 1.41	1.6 ± 0.21
HFD control	65.7 ± 12.90	46.0 ± 1.00	140.0 ± 9.54 ^#^	63.3 ± 2.89 ^#^	19.0 ± 1.00 ^#^	235.0 ± 37.98 ^#^	236.3 ± 50.06 ^#^	2.5 ± 0.15 ^#^
GE (mg/kg body weight/day)								
200	81.0 ± 9.85	52.0 ± 4.58	108.0 ± 5.29 **	58.0 ± 1.00	14.0 ± 1.00	186.0 ± 24.06	114.0 ± 9.17 **	2.4 ± 0.12
GGE (mg/kg body weight/day)								
50	72.7 ± 5.03	53.7 ± 3.06	100.0 ± 5.29 **	62.0 ± 2.65	12.0 ± 1.00 **	136.3 ± 8.96 **	97.3 ± 5.51 **	2.3 ± 0.10
100	71.0 ± 8.72	53.7 ± 3.06	104.7 ± 7.51 **	60.0 ± 2.00	14.3 ± 4.04	167.7 ± 31.34 *	103.0 ± 28.21 **	2.2 ± 0.10
200	71.5 ± 10.61	45.7 ± 5.86	97.0 ± 1.41 **	63.0 ± 4.39	15.0 ± 1.00	160.0 ± 14.00 **	130.0 ± 0.63 *	2.0 ± 0.10 **

Data are mean ± SD (each group, n = 6). ^#^ *p* < 0.05 vs. normal control, * *p* < 0.05, ** *p* < 0.01 vs. HFD control. HFD, high-fat diet; GE, EtOH extract of fresh ginger; GGE, EtOH extract of golden ginger, which is steam-processed ginger; HDL-C, high-density lipoprotein cholesterol; LDL-C, low-density lipoprotein cholesterol; GOT, aspartate aminotransferase; GPT, alanine aminotransferase; NEFA, non-esterified fatty acid.

**Table 5 ijms-25-02982-t005:** The primer sequences used in the quantitative RT-PCR.

Gene	Primer Sequences
Peroxisome proliferator-activated receptor γ (PPARγ)	5′-CGCTGATGCATGCCTATGA-3′ (sense) 5′-AGAGGTCCACAGAGCTGATTCC-3′ (antisense)
CCAAT/enhancer binding protein (C/EBPα)	5′-CGCAAGAGCCGAGATAAAGC-3′ (sense) 5′-CACGGCTCAGCTGTTCCA-3′ (antisense)
Adipocyte protein 2 (aP2)	5′-CATGGCCAAGCCCAACAT-3′ (sense) 5′-CGCCCAGTTTGAAGGTTCTCA-3′ (antisense)
Cluster of differentiation 36 (CD36)	5′-GCTTGCAACTGTCAGCACAT-3′ (sense) 5′-GCCTTGCTGTAGCCAAGAAC-3′ (antisense)
Acetyl CoA carboxylase (ACC)	5′-GAATCTCCTGGTGACAATGCTTATT-3′ (sense) 5′-GGTCTTGCTGAGTTGGGTTAGCT-3′ (antisense)
Fatty acid synthase (FAS)	5′-CTGAGATCCCAGCACTTCTTGA-3′ (sense) 5′-GCCTCCGAAGCCAAATGAG-3′ (antisense)
Liver X receptor (LXR)	5′-AGGCCGGTGCTGAGTATGTC-3′ (sense) 5′-GGGCTCCATAAAGTCACCAA-3′ (antisense)
Sterol regulatory element binding transcription factor-1c (SREBP-1c)	5′-GGCTCCTGCCTACAGCTTCT-3′ (sense) 5′-CAGCCAGTGGATCACCACA-3′ (antisense)
Glucose transporter type-4 (GLUT-4)	5′-AGAGTCTAAAGCGCCT-3′ (sense) 5′-CCGAGACCAACGTGAA-3′ (antisense)
Insulin receptor substrate 1 (IRS-1)	5′-GCCAATCTTCATCCAGTTGC-3′ (sense) 5′-CATCGTGAAGAAGGCATAGG-3′ (antisense)
Myostatin	5′-GGCCATGATCTTGCTGTAAC-3′ (sense) 5′-TTGGGTGCGATAATCCAGTC-3′ (antisense)
Myoblast determination protein (MyoD)	5′-GGCTACGACACCGCCTACTA-3′ (sense) 5′-GTGGAGATGCGCTCCACTAT-3′ (antisense)
*β*-actin	5′-TGTCCACCTTCCAGCAGATGT-3′ (sense) 5′-AGCTCAGTAACAGTCCGCCTAGA-3′ (antisense)

## Data Availability

The data presented in this study are available on request from the corresponding author.

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
