# Peer review of "The Effects of Body Fat Reduction through the Metabolic Control of Steam-Processed Ginger Extract in High-Fat-Diet-Fed Mice"

_ijms, 2024, doi:10.3390/ijms25052982_

Round 1

Reviewer 1 Report

Comments and Suggestions for Authors

There are some comments for this study as follows:

1. The author performed several experiments in this study, but their correlation was not elucidated clearly(e.g., the relationship between antioxidant properties, anti-glucosidase properties, and anti-obesity).

2. The author should provide the qualitative data of 1-dehydro-6-gingerdione in GE and GGE. Also, the author should address the reason for choosing this chemical as the bioactive compound of ginger in the present study. Finally, the author should include a treatment group of 1-dehydro-6-gingerdione in all experiments.

3. The specific effect of the steaming process on ginger quality should be discussed critically in the present study.

4. The author should draw a figure such as a graphical abstract to highlight the main findings of this study.

5. All figures and tables should be reorganized and re-formated. The bar chart should be revised to the box plot, and the SEM should be changed to SD.

Comments on the Quality of English Language

Moderate editing of the English language is required for the present study.

Author Response

  1. The author performed several experiments in this study, but their correlation was not elucidated clearly(e.g., the relationship between antioxidant properties, anti-glucosidase properties, and anti-obesity).

→ We thank the reviewer for the valuable feedback. As you mentioned, we have supplemented and revised the discussion section to further discuss the relationship between antioxidant properties, anti-glucosidase properties, and anti-obesity.

  1. The author should provide the qualitative data of 1-dehydro-6-gingerdione in GE and GGE. Also, the author should address the reason for choosing this chemical as the bioactive compound of ginger in the present study. Finally, the author should include a treatment group of 1-dehydro-6-gingerdione in all experiments.

→ Thank you for this valuable suggestion. As you mentioned, we attached the quantitative analysis data for 1-dehydro-6-gingerdione. The quantitative analysis confirmed that the content of this compound increased exponentially in the steam process and was set as a marker compound. In addition, in our previous study [Nam, Y. H., Hong, B. N., Rodriguez, I., Park, M. S., Jeong, S. Y., Lee, Y. G., ... & Kang, T. H. (2020). Steamed ginger may enhance insulin secretion through KATP channel closure in pancreatic β-cells potentially by increasing 1-dehydro-6-gingerdione content. Nutrients, 12(2), 324.], we demonstrated the efficacy of 1-dehydro-6-gingerdione in α-glucosidase inhibitory activity. The results related to inflammatory markers were included in the initial draft, but they were removed due to overlapping with our paper under review. We stated the efficacy of the compound in the manuscript. We hope this response addresses your valuable suggestion.

  1. The specific effect of the steaming process on ginger quality should be discussed critically in the present study.

→ Thank you. As you suggested, validation was performed to check the quality of GGE and we further described the efficacy of 1-dehydro-6-gingerdione produced in the steam process in the discussion section.

  1. The author should draw a figure such as a graphical abstract to highlight the main findings of this study.

→ A graphical abstract has been uploaded.

  1. All figures and tables should be reorganized and re-formatted. The bar chart should be revised to the box plot, and the SEM should be changed to SD.

→ Thank you. All figures and tables were revised as suggested.

Reviewer 2 Report

Comments and Suggestions for Authors

Author Response

The manuscript: "Steaming enhances the anti-obesity effects of ginger in high-fat-diet fed mice" represents a study that compare the anti-obesity effect between steam-processed ginger extract and conventional ginger extract. Even the manuscript includes various experimental systems for evaluation of anti-obesity activity of ginger extracts, it should be appropriately revised and more clearly presented. Thus, I recommend it to be published in the International Journal of Molecular Sciences after major revision of the manuscript

The following changes are recommended:

Modification in the Title

The title may be modified since it looks like steaming affects the obesity in mice. As an alternative, the authors can use: "The anti-obesity effects of steam-processed ginger extractin high-fat-diet fed mice" or something similar.

→ We thank the reviewer for the valuable feedback. As you suggested, we revised the manuscript title as follows: The effects of body fat reduction through metabolic control of steam-processed ginger extract in high-fat-diet-fed mice.

Modification in the section Abstract

Pg. 1, Line 18-19: The following sentence should be revised: "Among natural product, we focused on as anti-obesity candidate of Zingiber officinale rhizomes which have been used as a spice and functional food."

→ The sentence has been revised as suggested.

Modification in the section Introduction

The authors should give a brief review from previously published data for anti-obesity effects of ginger or its active compounds. Thereafter, the authors should make some introduction for potential biological activities of 1-dehydro-6-gingerdione, particularly anti-diabetic or anti-obesity effects. Are there any references for the bioactivity of this compound since the authors have been focused on its identification and isolation in ginger extracts. The novelty of the study should be emphasized.

→ Thank you for this valuable suggestion. We have revised the introduction section to emphasize our research.

Modification in the section Results

Figure 1: The GE and CGE should be provided in full names in Figure 1 Caption.

→ Thank you. The figure caption has been modified as suggested.

Table 1: The meaning of symbol "#" should be defined in Table 1 footnote.

→ The table footnote has been modified as suggested you.

Pg. 3, Line 106-107: The following statements should not be included in the section results: "α-glucosidase is a key enzyme in carbohydrate metabolism. α-glucosidase inhibitors, such as acarboses, can suppress postprandial hyperglycemia."

→ Thank you. The sentence has been moved to the discussion section as suggested.

Also the following statements should be included in Discussion section:

- Line 127-129: "Adipogenesis (adipocyte differentiation) is controlled by a transcriptional network coordinated by numerous transcription factors, including C/EBPα and PPARγ, which inhibit or promote adipocyte differentiation [2]."

- Line 153: "Inhibition of lipid accumulation in the liver may help treat nonalcoholic fatty liver."

- Line 185-186: "Measurement of glucose uptake by cultured myotubes is a reliable method to assess if an intervention can improve insulin responsiveness or correct insulin resistance [23]."

- Line 308-311: "Diabetic and obese conditions alter the structural, metabolic, and functional characteristics of skeletal muscle fibers, leading to loss of muscle [24]. Myostatin acts as a negative regulator of skeletal muscle mass and is frequently increased in obesity. In contrast, myoblast determination protein D (MyoD) plays a critical role in myogenesis."

- Line 508: "Oleic acid can induce excessive lipid deposition in liver cells [30]."

→ Thank you. These sentences were moved to the discussion section as suggested.

Pg. 4, Line 127-135: The results for the effect of GGE on transcript levels of C/EBPα, PPARγ, aP2, FAS and CD36 in 3T3-L1 cells should be more clearly presented.

→ The sentence has been rewritten as suggested.

Figure 2: The following statements from Figure 2 Caption should be included in the section Material and Methods: "During differentiation, the 3T3-L1 cells were treated in the presence or absence of GE and GGE for 8 days. At the end of the experiment, AdipoRed staining was performed to determine the rate of adipocyte differentiation. Intensities of fluorescence was measured with excitation and emission wavelengths at 485 nm and 535 nm, respectively." Similarly, check the Figure 3, 4 and 5 Captions.

→ Thank you. All the figure captions have been revised as suggested.

Figure 2: This figure should be better organized and presented since it is consisted of two subfigures 2A and 2B, while 2B is additionally subdivided into a,b,c,d and e. This should also be implemented for Figure 3 and Figure 4.

→ Thank you for this valuable indication. As the reviewer pointed out, Figs. 3 and 4 have been described separately. But in Fig. 2, we believe that it would be better to explain five genes at once in the manuscript. But in the figure, it is marked separately for convenience. We hope this response addresses your valuable suggestion.

Figure 3: On the graphs of this figure there is no sign "**" as indicated in Figure 3 Caption: "**; p<0.01 vs 5% oleic acid-treated control". Please, check it.

→ The figure caption has been revised as suggested.

Table 3 and Table 4: The data in these tables for each parameter in the groups cannot be clearly distinguished. Maybe, the Tables should be modified for better visualization of the results.

→ Thank you. Tables 3 and 4 have been revised as suggested.

Modification in the section Discussion

Pg. 11, Line 326-335: The first paragraph of the section Discussion looks like Conclusion, where authors summarized the main findings of the study. With some modification this part could be combined with the presented Conclusion section in the manuscript.

→ Thank you for this valuable indication. The whole conclusion section has been rewritten as suggested.

Pg. 11, Line 342-344: What is the meaning of "increased exponentially" in the following statement: "Likewise, in this study, it was confirmed that the content of 1-dehydro-6-gingerdione, a major component of this plant, increased exponentially in the steam process to GGE."? Did the authors dynamically evaluate the compounds’ content during the steaming?

→ In this experiment, the content analysis was performed before and after steaming as described in the manuscript. The term has been modified to avoid confusion (exponentially→dramatically) as suugested.

Several topics should be discussed:

- What is the reason for higher antioxidant activity of GGE compared to GE? Does 1-dehydro-6-gingerdione contribute to the antioxidant properties? Any explanation?

- What is the reason for higher alpha-glucosidase inhibitory activity of GGE compared to GE? What about the structure moieties of 1-dehydro-6-gingerdione that could be involved in the enzyme inhibition?

→ Thank you. The cause of increasing antioxidant capacity was revealed as major compound in GGE, 1-dehydro-6-gingerdione, which is produced during the steam process. Due to its "Keto-Enol Form", antioxidant capacity is enhanced [1,2]. In addition, as described in the description in the discussion part, antioxidant efficacy is related to alpha-glucosidase inhibitory activity. This information has been added in the discussion section, and its chemical structure is shown in the following references.

  1. Weber, W. M., Hunsaker, L. A., Abcouwer, S. F., Deck, L. M., & Vander Jagt, D. L. (2005). Anti-oxidant activities of curcumin and related enones. Bioorganic & medicinal chemistry, 13(11), 3811-3820.
  2. Litwinienko, G., & Ingold, K. U. (2004). Abnormal solvent effects on hydrogen atom abstraction. 2. Resolution of the curcumin antioxidant controversy. The role of sequential proton loss electron transfer. The Journal of organic chemistry, 69(18), 5888-5896.
  3. NORDIN, Nurul Izza. Immunomodulatory effects of Zingiber officinale Roscoe var. Rubrum (Halia Bara) on inflammatory responses relevant to psoriasis. 2012. PhD Thesis. Queen Mary University of London.

- Any explanation or discussion for the following statements:

o "GGE treatment increased mRNA expression levels of IRS-1 and GLUT4 in C2C12 cells to a greater extent than GE and increased 2-NDBG uptake more than GE"

o "Our results indicate that GGE supplementation would be better than GE supplementation at protecting against loss of muscle mass."?

→ Thank you. We have described additional explanation in the discussion section.

- How authors can connect the results for the effect of GGE on lipid accumulation in the liver, biochemical lipid status and expression of lipid accumulation-associated genes in the liver of the animals?

→ These relationships have been further discussed in the discussion section.

The main findings in this study are that GGE samples containing higher content of 1-dehydro-6-gingerdione showed better results compared to GG sample with small content of this compound. The Discussion section should be focused on the anti-obesity effects of this compound through the analyzed parameters. In addition, the authors did not discussed potential contribution of other ginger phytochemicals that may be present in the extracts to the anti-obesity activity. The present results should also be compared with other studies related to the anti-obesity activity of ginger extracts. Therefore, Discussion section should be significantly revised and improved

→ We appreciate the reviewer for the constructive and encouraging comments. We have worked hard to prepare for revised manuscript incorporating your valuable suggestions. We hope that we have addressed your valuable comments.

Round 2

Reviewer 1 Report

Comments and Suggestions for Authors

The manuscript was revised well, but the labels on each figure should be consistent. The marks (a) and (b) of Figure 4B and the marks of Figures 6 and 7 should be moved to the left side. Also, there is no figure summarizing the findings of the present study in this manuscript.

Author Response

We sincerely appreciate the valuable feedback from the reviewer. Following your suggestion, we have ensured consistent labeling on each figure. Additionally, we have attached the previously submitted graphic abstract, which summarizes the findings of the study.

Reviewer 2 Report

Comments and Suggestions for Authors

Discussion section could be additionally improved according to the suggestions. 

Author Response

We sincerely appreciate the valuable feedback provided by the reviewer. Taking into account the suggestions in both the current and previous comments, we have further revised the discussion section to best enhance our manuscript.